# Technological Solutions for Sustainable Development: Effects of a Visual Prompt Scaffolding-Based Virtual Reality Approach on EFL Learners' Reading Comprehension, Learning Attitude, Motivation, and Anxiety

**Zhiqiang Wang** [1], **Yu Guo** [2], **Yan Wang** [3], **Yun-Fang Tu** [4] and **Chenchen Liu** [2,*]

1   Department of Educational Science, Wenzhou University, Wenzhou 325035, China; wangzhiqiang226@126.com
2   Department of Educational Technology, Wenzhou University, Wenzhou 325035, China; wzugy6@126.com
3   School of Foreign Languages, Wenzhou University of Technology, Wenzhou 325035, China; wangyanwzut@163.com
4   Department of Library and Information Science, Research and Development Center for Physical Education, Health, and Information Technology, Fu Jen Catholic University, New Taipei City 24205, Taiwan; sandy0692@gmail.com
*   Correspondence: lcc5424548@126.com

**Abstract:** As is indicated by the United Nation's Sustainable Development Goal 4, it is crucial to have access to inclusive and quality education for all. For English as a Foreign Language (EFL) learners, reading English is a basic skill for learners to acquire and exchange information and to have lifelong learning experiences. To provide a vivid EFL learning environment, a visual prompt scaffolding-based VR (VPS-VR) approach was proposed to enhance students' reading comprehension skills. To investigate the effectiveness of the proposed approach, an experiment was conducted in an English reading course at a Chinese university. Students from experimental group A (*N* = 31) learned with the VPS-VR approach, experimental group B (*N* = 32) learned with the virtual reality (VR) approach, and the control group learned with the traditional instruction (TI) approach. The results revealed the positive effects of the VPS-VR approach on students' EFL reading comprehension, learning motivation, and English learning anxiety. Furthermore, it was also found that experimental students' lower-level skills of reading comprehension, such as information location and text comprehension, were significantly improved, rather than the higher-level skills of reflection and evaluation. Fifteen students participated in interviews, and their learning experience and technology acceptance are also discussed.

**Keywords:** sustainable learning ability; VR; visual prompt scaffolding; EFL reading

## 1. Introduction

In light of recent globalization, scholars have repeatedly indicated that it is important to equip learners with an awareness of lifelong learning in a sustainable development society [1,2]. As indicated by Sustainable Development Goal 4, it is necessary to provide equitable access to quality education and lifelong learning opportunities for everyone [3]. In line with the trend of lifelong learning for all in a sustainable development society, learners are expected to be equipped with the awareness, skills, and strategies to have access to, process, and exchange information efficiently and effectively [4]. Hulme and Snowling [5] stressed the importance of reading strategy awareness for improving learners' academic achievements. Scholars have also pointed out that reading is a complex process in which learners have to integrate their perceptual, lexical, and processing skills to translate, recognize, and identify what they read [6]. Therefore, appropriate reading strategies are helpful for learners to repair deficits in their understanding of the text and to enhance

their learning performance [7]. With regard to EFL learning, reading comprehension in English is a basic and essential skill for students, since reading is the first step towards promoting students' acquisition and exchange of information [8–10]. Deficiency in English as a Foreign Language (EFL) reading comprehension is a major problem for most students in the development of their EFL skills [11]. With the development of ICT and diverse digital devices, digital reading requires students to deal with multimodal information with a high level of information integrating skills in reading activities, including seeing, perceiving, understanding, vocalizing, and mentally constructing [12,13]. However, the majority of EFL instructors still focus on correcting learners' grammar or increasing their vocabulary size [14], and fewer strategic or instrumental tools are provided for students to facilitate their reading comprehension. In the field of EFL, insufficient reading comprehension is often caused by new vocabulary and a lack of an authentic language environment to support understanding. It is suggested that integrating technology into EFL teaching and learning would innovate pedagogy and improve learning quality.

EFL teaching and learning approaches usually emphasize that learning occurs through intensive interaction and maximum use of the target language [15]. Recently, there has been an increasing number of studies attempting to integrate technologies to assist language learning by providing interactive and experience-based learning environments to promote the effects of EFL learning [4,16–20]. Jamshidifarsani et al. [21] analyzed different teaching methods and technologies to support EFL reading comprehension, such as web-based [22], computer-based [23], mobile device-based [24], and flipped classroom-based [25] EFL reading comprehension. However, the mentioned technology-assisted reading activities were still not able to provide authentic or in-field learning environments for EFL learners. VR provides an opportunity to cope with the problem of lacking an authentic learning context and social presence to create a realistic learning experience to enhance students' learning achievement [26].

Virtual reality provides immersive and virtual environments for students to experience situations that are not easily accessed physically due to time constraints, physical inaccessibility, or dangers in real situations [27]. In addition, VR technology has the potential to increase learning motivation, and therefore, has a positive impact on learning performance [28]. From this perspective, most EFL students could benefit from virtual learning environments to experience authentic foreign language conversation with the support of Head-Mounted Display VR or desktop-based VR [29]. Over the past few decades, various studies have explored the educational advantages and potential of VR in foreign language teaching and learning, for example, Ebadi and Ebadijalal's [30] findings on the impact of Google Expeditions VR on EFL learners' oral proficiency. Xie et al. [31] used mobile-based VR to enhance learners' oral proficiency. Additionally, scholars have integrated spherical video-based VR to improve students' speaking performance [32]. Previous studies have introduced VR into EFL learning to improve learners' oral proficiency due to the advantages of VR in providing realistic learning experiences; however, investigations of the impact of VR in EFL reading are scarce. To the best of our knowledge, existing studies using VR to enhance learners' EFL reading either focused on vocabulary acquisition as a basis for EFL reading [33,34] or compared the differences in learning performance between immersive (with VR) and non-immersive (without VR) learning [35,36]. Investigating EFL reading behavior and performance in immersive learning environments with the help of VR technology is, therefore, severely lacking and much needed. Furthermore, immersive and authentic learning experiences might arouse students' interest in reading activities, thereby improving their EFL reading comprehension performance.

Reading in immersive learning environments usually requires learners to adapt to a multimodal learning environment. Thus, it is necessary to provide appropriate scaffolding strategies during the reading process [37]. One of the scaffolding strategies often used in technology-enhanced learning environments is prompts, which can enhance metacognitive planning and reflection and improve content understanding [38]. Visual prompt scaffolding (VPS) in technology-enhanced learning environments generally involves highlighting,

labeling, and commenting on aspects of visual representations to help focus learners' attention on specific learning aspects [39]. Therefore, the current study aimed to investigate the effects of the visual prompt scaffolding-based VR (VPS-VR) approach on EFL learners' reading comprehension. The research questions (RQ) were as follows:

RQ1: Can students who learn with the VPS-VR approach improve their EFL reading comprehension compared to students learning with the VR and TI approaches?

RQ2: Can students who learn with the VPS-VR approach improve their learning motivation and attitude in EFL reading comprehension compared to students learning with the VR and TI approaches?

RQ3: Can students who learn with the VPS-VR approach decrease their learning anxiety in EFL reading comprehension compared to students learning with the VR and TI approaches?

RQ4: What are the perspectives of the students learning EFL reading with the VPS-VR, VR, and TI approaches?

## 2. Literature Review

### 2.1. Immersive Learning Experience in VR

According to the embodied cognition theory, cognitive processes are grounded in sensory-motor experience [40]. Thus, multimodal stimulation provided by immersive learning environments can improve student learning performance. A powerful tool to implement a multisensory yet controllable experience is fully immersive VR, which is typically experienced by Head-Mounted Display or Cardboard to provide realistic environments in which users can interact with virtual objects to simulate the experience of interactions in the real world [41]. Researchers have indicated the potential of integrating VR into the education field to positively affect learning performance and higher order thinking [42]. In the previous study, VR technology has been applied in diverse educational contexts, including STEM-related disciplines, such as physics [43], chemistry [44], medical courses [45], and even art and history [46]. From the application domains of VR, it can be indicated that the main advantages of VR technology in the educational context are that it provides practice opportunities without time and space limitations and simulated learning environments for dangerous learning units [47]. Despite the advantages of VR technology mentioned above, teachers in schools have still reported that the high cost and difficulty of editing VR learning material hindered them from introducing VR in daily pedagogy [48]. Spherical video-based virtual reality (SVVR), a form of low cost and easy-to-implement VR, seems to be the resolution and has the potential to be introduced in daily teaching and learning activities. It uses immersive video content, allows users to look around in all directions, and provides them with the opportunity to control what they want to see [32]. Moreover, SVVR can explore immersive learning environments from the first-person perspective to provide learning content through natural movements to create a powerful embodied experience [49].

### 2.2. VR in EFL Teaching and Learning

Due to time and space limitations, most EFL learners do not have the opportunity to practice, reflect, and interact with their English in a meaningful context [48]. Consequently, VR is usually regarded as a potential approach to create meaningful and authentic contexts and improve students' learning motivation and engagement in foreign language learning [49]. In the past few years, scholars have investigated the effect of introducing VR in different disciplines of EFL, such as oral efficiency [31,32], listening [50], and writing [51]. However, studies on the effects of VR in EFL reading either investigate its impact on learners' EFL vocabulary acquisition as a foundation for reading [33,52], or focus on the impacts of different technologies (e.g., VR, AR, and traditional 2D [53]. Thus, studies investigating EFL reading performance and behavior and appropriately integrating visual prompt scaffolding strategies are needed.

### 2.3. EFL Reading Comprehension in Immersive Learning Environments

In general, there have been different definitions of EFL reading comprehension in digital learning environments, since researchers often define the concept according to their own research field. A widely accepted definition is the cognitive process of transforming texts, graphics, and other multimodal stimuli into information to understand [54]. In addition, scholars have pointed out that it is necessary to consider the ubiquities of EFL reading comprehension in comparison with the mother language, namely, prior literacy experience in the reading process, limited sophistication, and dual-language involvement [7,9]. In the past decades, researchers have explored the introduction of different technologies to enhance EFL reading comprehension, including computer-based [55], mobile device-supported [56], and AR-assisted [57] learning environments. However, scholars have indicated that the effectiveness of integrating diverse instruments (immersive or non-immersive) highly depends on the instructional design and the appropriateness of the learning strategies adopted. Previous studies have integrated diverse strategies, including game-based learning [33], peer assessment [32], concept mapping [55], collaboration annotation [58], and EFL reading activities. Among these strategies, annotation is a common method used to remind students to pay attention to the key elements in the reading text or structured reading content [59]. There are many types of annotations, including text highlighting, explanatory information, and knowledge extension in the digital reading process [60], which are known to be effective in terms of improving comprehension [61]. In immersive learning environments (e.g., VR), reading comprehension is challenging; however, it has not yet been explored. Therefore, the purpose of this study is to investigate the effectiveness of the VPS-VR approach on students' EFL reading comprehension performance and perceptions.

### 2.4. The VPS-VR Approach for EFL Reading Comprehension

In this study, a learning environment with the VPS-VR approach for EFL reading comprehension was implemented with the support of the editing instruments of Adobe Premiere Pro. Figure 1 shows the system structure of the learning environments. Teachers were able to edit the 3D video-based VR to provide VPS for learners during reading in the VR video, while students were able to watch, learn, and reflect in the learning environments.

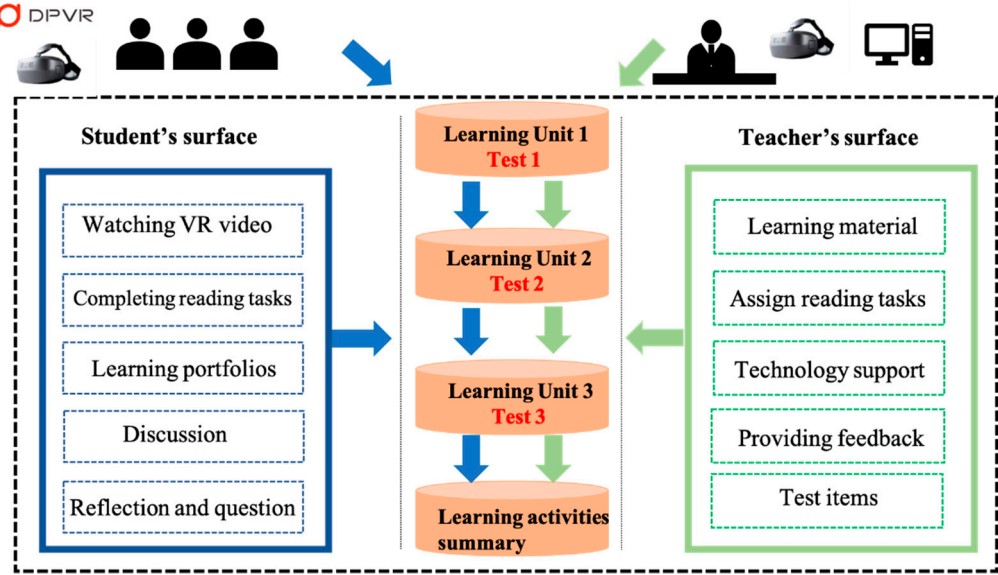

**Figure 1.** Introduction of the learning system structure.

The device adopted in this study is the M2 Pro of DPVR. This device is compatible with the M-Polaris system, which enables free movement within the VR scene and accurately captures the user's position and movement. The device is portable and easy to operate with the

help of a touchpad and physical buttons. In addition, there is an object distance adjustment button on the top of the device, which is convenient for students with short-sightedness.

The reading resources in the current study originated from the TV series SpongeBob SquarePants and were edited by teachers to organize the reading activities in the VR environments. The learning content was based on three different spherical videos, namely, Recipe for Krabby Patties, Scavenger Hunt, and Patrick's Overpayment. Students watched the videos from the first-person perspective and were required to complete the reading comprehension test at the end of the video. According to the PISA test [62], reading comprehension included three aspects of reading comprehension skills: information location, text comprehension, and evaluation and reflection. As shown in Figure 2, learners in experimental group A watched the VR video with VPS strategies, experimental group B watched the VR video without VPS strategies, and the control group watched the TI approach. The learning content of the three groups was identical.

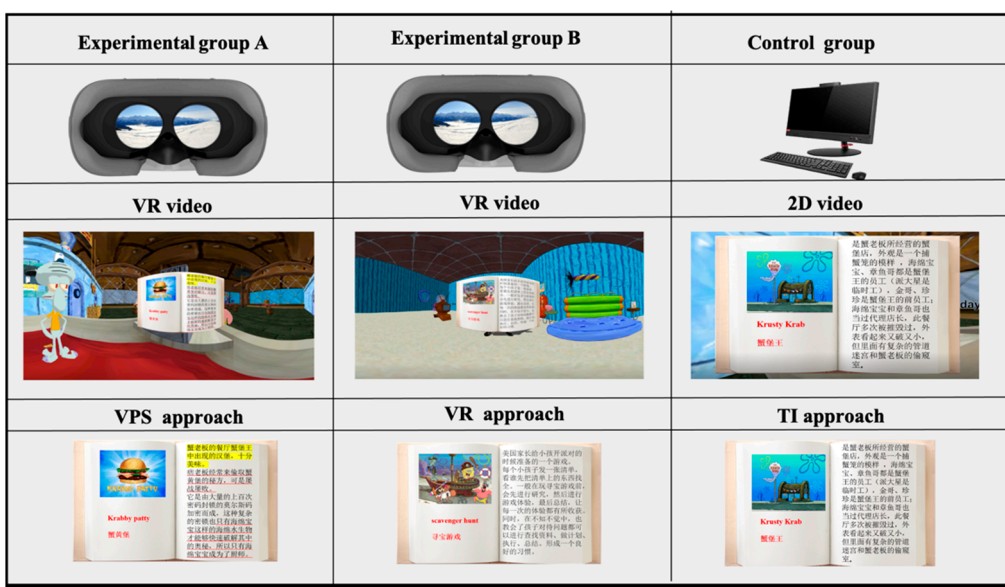

**Figure 2.** Learning environments of experimental group A, experimental group B, and the control group.

The VPS strategies adopted in the current study provided explanatory and supplementary information while reading in the VR environments, as shown in Figure 3. Students were required to watch the 3D video-based VR and to read real-time subtitles as well as explanatory information in order to complete the reading comprehension tests. For example, when a new word came up in the VR video, a flashcard with the explanatory information showed up for six to eight seconds and then disappeared, and then students continued to watch the video.

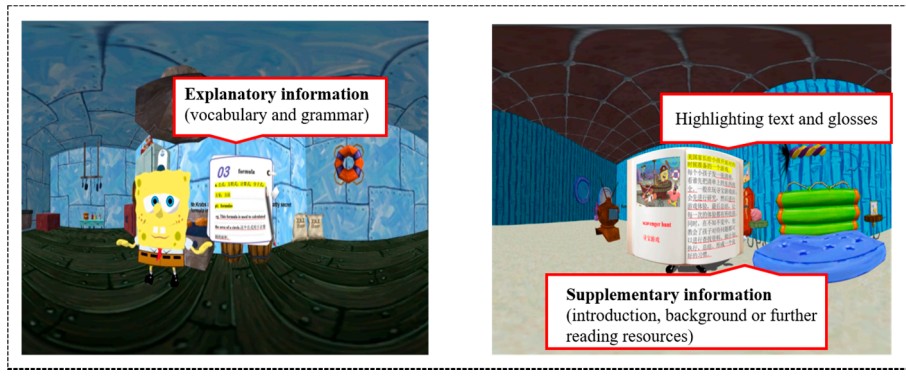

**Figure 3.** System structure of learning with the VPS-VR approach.

## 3. Method

### 3.1. Research Design

In this study, three different learning approaches (VPS-VR, VR, and TI) in the EFL reading learning activities were defined as independent variables. The post-test and post-test questionnaires for EFL reading comprehension, learning motivation, learning attitude, and English learning anxiety were the dependent variables, while the pre-test and pre-test questionnaires were the covariates.

### 3.2. Participants

The study used a quasi-experimental design. The participants were 98 college students from three classes taught by the same instructor. Their average age was 19. One class was experimental group A (*N* = 31) learning with the VPS-VR approach, one class was experimental group B (*N* = 32) learning with the VR approach, and the other class was the control group (*N* = 35) learning with the TI approach.

All the students had previous experience of digital reading, but none had any experience using VR in English courses. The participants' anonymity was protected by concealing their personal information during the research process. They knew that their participation was voluntary and that they could withdraw from the study at any time.

### 3.3. Experimental Procedure

The experiment was conducted at a Chinese university. Before starting the experiment, the teacher introduced how to use VR devices for each group of students. A pre-test on students' English reading comprehension skills was provided by the teacher. Thereafter, the students were required to complete a learning attitude questionnaire. As shown in Figure 4, the experiment lasted for 3 weeks, and the reading activities in the VR environments lasted for 20 min per session. The three groups all needed to watch the video and complete the reading comprehension test according to the video.

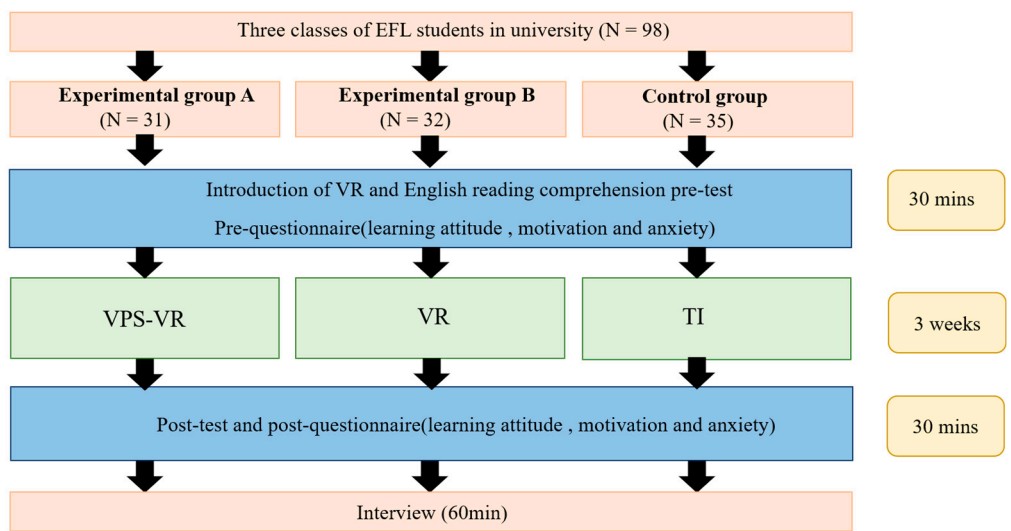

**Figure 4.** Experimental procedure.

It should be noted that the students in experimental group A watched a VPS-VR video, experimental group B watched a VR video, and the students in the control group were shown the same content via the TI approach. At the end of the 3-week experiment, students completed post-test questionnaires on motivation, learning attitudes, and English learning anxiety. Fifteen students participated in the interviews to investigate their learning experience.

## 4. Instruments

### 4.1. English Reading Comprehension Test

To gauge the participants' prior reading performance, a reading performance pre-test was administered by two experienced teachers. In addition, three experts in the field of EFL reading education were invited to check the reliability of the test. Learning performance was gauged from the participants' test scores in the reading post-tests. These post-tests were selected based on the three learning materials, which consisted of three dimensions: text comprehension, evaluation, and reflection. The two tests comprised 15 questions each, and each correct answer was awarded 2 points. Each test took approximately 30 min to complete.

### 4.2. Learning Motivation

The learning motivation questionnaire was developed based on the measure proposed by Pintrich et al. [63]. It consists of three items with a 5-point Likert rating scale. An example item is, 'In a class like this, I prefer course material that really challenges me so I can learn new things'. Responses were given on an evaluation scale of five grades as follows: 1 = totally disagree, 2 = disagree, 3 = not sure, 4 = agree, and 5 = exactly agree. The questionnaire has a Cronbach's alpha of 0.79.

### 4.3. Learning Attitude

The learning attitude questionnaire was modified from that developed by Hwang et al. [64] and adopted a 5-point Likert scale. An example item is, 'I find learning this course interesting and rewarding'. Responses were given on an evaluation scale of five grades as follows: 1 = totally disagree, 2 = disagree, 3 = not sure, 4 = agree, and 5 = exactly agree. The questionnaire has a Cronbach's alpha of 0.78.

### 4.4. English Learning Anxiety

The English learning anxiety questionnaire by Horwitz et al. [65] consists of 33 questions and uses a 5-point Likert scale. An example item is, 'When I speak in foreign language classes, I am never sure of myself'. Higher ratings represent higher anxiety about learning English. Responses were given on an evaluation scale of five grades as follows: 1 = totally disagree, 2 = disagree, 3 = not sure, 4 = agree, and 5 = exactly agree. The Cronbach's alpha of the measure was 0.94.

### 4.5. Interview

The interview questions in this study were modified from the measure developed by Hwang et al. [66] (see Appendix A). It consists of nine questions to collect participants' perceptions of the learning activity, and all participants were marked according to E01-E05, CA01-CA05, and CB01-CB05. The interview content was recorded as audio files for analysis. Examples of the interview questions include: 'What is the advantage of this learning approach that benefits you the most? Why do you think so? Please give an example' and 'What's the difference between the experience in this learning activity and your previous learning experience?'.

## 5. Experimental results

### 5.1. Analysis of Reading Performance

After conducting the learning activity, ANCOVA was performed on the post-test results, in which the pre-test was the covariant, the post-test results were the dependent variable, and the different learning methods (three groups) were the control variable, to compare the reading performance of the three groups. As is shown in Table 1, the ANCOVA results showed that the difference between the three groups was significant ($F = 4.003$, $p < 0.05$) after the impact of the pre-test scores on the post-test was excluded, implying that the post-test scores of the three groups were significantly different due to the different experimental learning processes. Furthermore, post hoc analysis was performed to examine



specific differences in EFL reading comprehension between the three groups, with adjusted means of 18.070, 15.105, and 17.960, respectively. The least significant difference (LSD) test revealed that the reading scores of experimental group A (the VPS-VR approach) were significantly higher than those of experimental group B (the VR approach). Accordingly, it was concluded that in VR learning environments, students using the VPS strategy had better reading performance than those without the VPS strategy.

**Table 1.** The results of the ANCOVAs on students' reading performance.

| Group | *N* | Mean | *SD* | Adjust Mean | *F* | Post hoc (LSD) |
|---|---|---|---|---|---|---|
| Experimental group A | 31 | 18.06 | 5.477 | 18.070 | 4.003 * | (1) > (3) |
| Experimental group B | 31 | 15.35 | 4.513 | 15.105 | | |
| Control group | 35 | 17.74 | 4.182 | 17.960 | | |

* $p < 0.05$.

To further investigate the results, this study analyzed students' reading scores in terms of three dimensions: information location, text comprehension, and evaluation and reflection, through ANCOVA, as shown in Table 2. The reading scores of the three groups differed significantly in the dimensions of information location ($F = 7.996$, $p < 0.01$) and text comprehension ($F = 14.062$, $p < 0.001$), but not in the dimension of evaluation and reflection ($F = 0.043$, $p > 0.05$). In particular, in the information location dimension, students learning with the VPS-VR approach (adjusted mean of 8.610 for experimental group A) scored higher than those learning with the VR approach (adjusted mean of 6.778 for experimental group B; adjusted mean of 6.656 for the control group). In the text comprehension dimension, students learning with the TI approach (adjusted mean of 7.707 for the control group) performed better than those learning with the VR approach (adjusted mean of 5.851 for experimental group A; adjusted mean of 4.835 for experimental group B). This result answered the RQ1, that is, the students learning with the VPS-VR approach improved their EFL reading comprehension more than those learning with the VR and TI approaches.

**Table 2.** ANCOVA results analyzing the scores of reading performance by the five dimensions.

| Dimensions | Group | *N* | Mean | *SD* | Adjust Mean | *F* | $\eta^2$ |
|---|---|---|---|---|---|---|---|
| Information location | Experimental group A | 31 | 8.71 | 2.807 | 8.610 | 7.996 ** | 0.147 |
| | Experimental group B | 31 | 6.84 | 1.917 | 6.778 | | |
| | Control group | 35 | 6.51 | 1.704 | 6.656 | | |
| Text comprehension | Experimental group A | 31 | 5.74 | 2.352 | 5.851 | 14.062 *** | 0.232 |
| | Experimental group B | 31 | 4.97 | 2.121 | 4.835 | | |
| | Control group | 35 | 7.69 | 2.259 | 7.707 | | |
| Evaluation and reflection | Experimental group A | 31 | 3.55 | 1.234 | 3.550 | 0.043 | 0.001 |
| | Experimental group B | 31 | 3.48 | 1.363 | 3.473 | | |
| | Control group | 35 | 3.54 | 1.094 | 3.550 | | |
| Overall | Experimental group A | 31 | 18.06 | 5.477 | 18.070 | 4.003 * | 0.079 |
| | Experimental group B | 31 | 15.35 | 4.513 | 15.105 | | |
| | Control group | 35 | 17.74 | 4.182 | 17.960 | | |

* $p < 0.05$ ** $p < 0.01$ *** $p < 0.001$.

### 5.2. Analysis of Learning Motivation

To examine the effect of the three learning methods on students' learning motivation, ANCOVA was performed on the post-test results, in which the learning motivation pre-test was the covariant, the post-test results were the dependent variable, and the different learning methods (three groups) were the control variable, to analyze the learning motivation of the three groups. The ANCOVA results showed (Table 3) that there were significant differences in learning motivation among the three groups ($F = 3.483$, $p < 0.5$). Furthermore, post hoc analysis was performed to examine the specific differences in the achievements

of the three groups. The least significant difference (LSD) test revealed that the learning motivation of experimental group A (the adjusted average was 24.468) was significantly higher than that of experimental group B (the adjusted average was 23.405) and the control group (the adjusted average was 23.427), and there was no significant difference in learning motivation between experimental group B and the control group. It was concluded that the VPS-VR approach can improve students' learning motivation. This result answered the RQ2, that is, the students learning with the VPS-VR approach improved their learning motivation in EFL reading comprehension more than those learning with the VR and TI approaches.

**Table 3.** The results of the ANCOVAs on students' learning motivation.

| Group | N | Mean | SD | Adjusted Mean | F | Post hoc (LSD) |
|---|---|---|---|---|---|---|
| Experimental group A | 31 | 25.48 | 3.345 | 24.468 | 3.483 * | (1) > (2) |
| Experimental group B | 31 | 23.19 | 2.960 | 23.405 | | (1) > (3) |
| Control group | 35 | 22.71 | 3.121 | 23.427 | | |

* $p < 0.05$.

### 5.3. Analysis of Learning Anxiety

To examine the effect of the three learning methods on students' learning anxiety, ANCOVA was performed on the post-test results, in which the learning anxiety pre-test was the covariant, the post-test results were the dependent variable, and the different learning methods (three groups) were the control variable, to analyze the learning anxiety of the three groups. The ANCOVA results showed that the difference between the three groups was significant ($F = 5.979$, $p < 0.01$) (Table 4). Furthermore, post hoc analysis was performed to examine the specific differences in achievement of the three groups. The least significant difference (LSD) test revealed that the learning anxiety of the control group (the adjusted average was 99.085) was significantly higher than that of experimental group A (the adjusted average was 96.427); there was no significant difference in learning anxiety between experimental group A and experimental group B, and the control group. It was concluded that word cards and extended reading were more likely to cause learning anxiety when they were unmarked, especially in a VR learning environment (the adjusted mean for experimental group B was 102.089). This result answered the RQ3, that is, the students learning with the VPS-VR approach had less learning anxiety in EFL reading comprehension learning, compared with those learning with TI approaches.

**Table 4.** The results of the ANCOVAs on students' learning anxiety.

| Group | N | Mean | SD | Adjust Mean | F | Post hoc (LSD) |
|---|---|---|---|---|---|---|
| Experimental group A | 31 | 98.45 | 17.767 | 96.427 | 5.979 ** | (3) > (1) |
| Experimental group B | 31 | 99.77 | 8.421 | 102.089 | | |
| Control group | 35 | 99.34 | 10.195 | 99.085 | | |

** $p < 0.01$.

### 5.4. Interview

Following Hwang et al. [64], the coding scheme was adapted for three aspects: learning experience, learning performance, and technology acceptance. The results of the coding are shown in Table 5, where N represents the frequency of each item mentioned by the students in the interview. We found that VR was more likely to enhance students' sense of learning experience ($N = 71$) and provide an 'immersive learning experience' and 'a variety of learning resources'. For example, E03 said, 'It brings a visual impact on learning that is more intuitive than books and PowerPoint because you are immersed in it and know you are in the environment'. E04 said that he 'prefers virtual reality technology for learning because it is acceptable, relatively new, and incorporates some colorful cartoon elements'. In addition, VPS strategies provide assistance for reading activities in virtual

environments that reduce learner anxiety, improve attention to learning, and provide practice in understanding (*N* = 82). For example, CA02 mentioned that 'it allows you to further deepen your impressions of words as you learn, and the biggest benefit is that it allows you to experience the context of speech.' CB04 said, 'It engages all five of our senses and makes us feel the joy of learning English'. The results of the technology usefulness and technology ease of use interviews revealed that the three groups of students had a high level of technology acceptance and benefited from technology-assisted learning (*N* = 154). For example, E03 said, 'I think I learned it quickly, like the words in SpongeBob SquarePants, the words that appear can be combined with the content of the video, it's not a single memory, it's interrelated'. CB03 noted, 'Words in context that can be remembered immediately and can be integrated into the context to feel the expansion of acceptance and receptivity, staring at the textbook feels inefficient'. E01 said, 'No difficulties, the operation is good and the system is very mature. The results of the interview answered the RQ4, indicating that students learning with the VPS-VR approach reported positive learning experiences, such as better learning experiences and a friendly learning environment, compared with those learning with the VR and TI approaches.

**Table 5.** The results of the interview.

| | Experimental Group B | | Control Group |
| --- | :---: | :---: | :---: |
| Improves learning experience | | | |
| Yes | | | |
| Immersive learning experience | 12 | 10 | 0 |
| Novel ways of learning | 8 | 11 | 0 |
| Learning environments are more authentic | 10 | 9 | 0 |
| Learning content is more visualised | 11 | 13 | 0 |
| No | | | |
| Causes discomfort | 2 | 2 | 0 |
| Lack of diversity | 2 | 1 | 1 |
| Enhances learning performance | | | |
| Yes | | | |
| Authentic learning environment | 10 | 12 | 2 |
| More concentrated on learning | 12 | 14 | 5 |
| Enhances interest or enjoyment of learning | 18 | 16 | 7 |
| Perceived usefulness | | | |
| Yes | | | |
| Speeds up word memorisation | 9 | 8 | 3 |
| Deepens knowledge understanding | 11 | 10 | 3 |
| Easier to understand | 10 | 9 | 8 |
| Focuses attention | 14 | 13 | 2 |
| No | | | |
| Reading difficulties: Short or unclear | 2 | 2 | 5 |
| Insufficient learning impressions | 1 | 2 | 6 |
| Perceived usability | | | |
| Easy to use | 16 | 13 | 0 |
| Mobility: Flexibility and convenience in learning | 11 | 9 | 5 |
| Technical problems | 3 | 2 | 0 |

## 6. Discussion

Since reading skills and strategies in the field of EFL learning are key drivers to achieve the aim of the UN's Sustainable Development Goal 4, 'achieving inclusive and quality education for all', the current study investigated the role of emergent technology in EFL reading comprehension. This study aimed to explore whether the VPS-VR approach could improve EFL learners' reading comprehension performance, learning motivation, and attitude, and reduce their EFL learning anxiety. In general, VR technology has often been adopted to enhance EFL listening and speaking, which creates authentic and innovative learning environments for EFL learners. Thus, this study expands the application domains of VR in EFL reading comprehension. In addition, this study also found that

appropriately integrating VPS into a VR learning environment is helpful for enhancing EFL reading comprehension.

Previous studies investigating the effects of VR with other teaching approaches [26,31,32] have indicated the potential of VR technology in allowing learners to access simulated, immersive, and interactive virtual environments to perform authentic learning activities [29]. Compared with other technology, such as computers [22], mobile phones [16], or tablets [66], VR technology has advantages in specific learning environments which may be dangerous or difficult to access [33]. However, it should be noted that VR implementation is costly in terms of preparing the learning material and designing the learning activities [46]. Besides, learners may easily feel dizzy or uncomfortable when wearing VR devices. Learning behaviors and interactions with others are also difficult to record.

According to the experimental results, students who learned with the VPS-VR approach performed statistically significantly better than those who learned with the TI approach in terms of EFL reading comprehension. The research results also indicated that students learning with the VPS-VR approach significantly improved their learning motivation and attitude in EFL reading comprehension compared with students who learned with the VR approach. It was also found that students learning with the VR approach reduced their EFL learning anxiety compared with those learning with the TI approach. The results of the current study are similar to those of Khezrlou et al. [34], who also found that VPS strategies, such as computer-based glosses in reading, enhanced students' comprehension and vocabulary acquisition. This research result identified the importance of appropriate reading strategies for improving learners' performance to ensure that students will not be excluded at the start of the learning process because of a lack of basic reading skills, which is also a point emphasized by Sustainable Development Goal 4.

To further investigate the effects of the VPS-VR approach on learners' reading comprehension performance, the current study also analyzed its impact on three aspects of reading comprehension, namely, information locating, text comprehension, and evaluation and reflection. It was found that students learning with the VPS-VR approach performed better than the other two groups in terms of information location and text comprehension rather than evaluation and reflection. This result indicated that the VPS strategies in the current study only improved the lower level of reading comprehension skills, including locating information and understanding unknown vocabulary. This is similar to a previous study by Chien et al. [32], who also found that the learning approach with VR did not significantly affect every aspect of students' learning behavior. Thus, other strategies to improve higher level reading comprehension skills need to be investigated.

Regarding motivation and learning attitude, students learning with the VPS-VR approach had significantly higher motivation and better attitudes compared to the control group. This result is similar to previous findings, which also adopted VR for foreign language learning, finding a positive impact on learning attitude [67]. Moreover, according to the interview results, they were provided with authentic learning environments, and some explanatory information helped them understand the reading content; thus, they felt confident about reading and the exercise that followed. In other words, with VPS strategies, students are more confident about reading and more efficient in learning in immersive learning environments. The results of this study are similar to those of past studies where scholars pointed out the advantages of VR technology in providing an authentic learning environment to improve learners' learning interests [33].

In terms of EFL learning anxiety, students learning with the VPS-VR approach had significantly lower EFL learning anxiety than those learning with other approaches. Consistent with the research results of Chien et al. [32], a learning approach supported by VR technology was found to be effective in terms of reducing English learning anxiety. In general, learners felt anxious about EFL learning activities when they received negative feedback [24]. Some students in the interview said that they felt free and safe when learning in VR learning environments since they did not need to worry if others' comments

were positive or negative. It can be inferred that immersive learning environments with appropriate VPS could enhance EFL learning by reducing learning anxiety.

However, there also exist several limitations in the current study, namely, the time limitation, limited learning resources, and lack of interaction in VR learning environments. First, this study lasted for only 3 weeks, whereas EFL reading comprehension is an ability that requires a long period of time to improve. Although the research results found significant differences between the experimental groups and control groups in two aspects of reading comprehension skills, a longer intervention is still needed to improve learners' reflection and evaluation of reading comprehension skills. Second, the learning resources of the 3D video-based VR were limited; more EFL learning resources could be explored in the future. In addition, the current study adopted 3D videos to train students in EFL reading comprehension skills, which did not provide them with opportunities to interact in the virtual learning environments. Thus, designing a VR learning environment with interactive functions could also be a direction for future research.

## 7. Conclusions

This research focused on scaffolding strategies in the EFL class that adopted VR technology. An experiment was conducted in a Chinese university. The aim of the current study was to investigate the effects of the VPS-VR approach on EFL learners' reading comprehension. To achieve this goal, an English reading comprehension test, a questionnaire of learning motivation and English learning anxiety, and a semi-structured interview were conducted. Significant differences were found in terms of EFL students' reading comprehension, learning motivation, and reduced English learning anxiety. Through the interview with students, it was also found that most of the students enjoyed the VR learning environments and benefited from the learning activities with the support of the VPS-VR approach. However, it should be noted that referring to reading skills, students' lower-level skills of reading comprehension, such as information location and text comprehension, were significantly improved, rather than the higher-level skills of reflection and evaluation. Therefore, further investigations of appropriate learning strategies integrated into VR learning environments are expected to improve students' higher-order thinking skills.

**Author Contributions:** Conceptualization, C.L.; Data curation, Y.W.; methodology, Y.G.; writing—original draft preparation, Z.W., and C.L.; writing—review and editing, Y.-F.T.; supervision, Z.W. and Y.-F.T.; project administration, Y.G., Y.W. and C.L.; funding acquisition, Z.W. All authors have read and agreed to the published version of the manuscript.

**Funding:** This research was funded by Talent Cultivating Project from Zhejiang Federation of Humanities and Social Sciences in China under contract number 21QNYC19d and Zhejiang Federation of Humanities and Social Sciences in China under contract number 22NDQN271YB.

**Institutional Review Board Statement:** All participants (98 college students) gave their informed consent for inclusion before they partici pated in the study. The study was conducted in accordance with the Declaration of Helsinki, and the protocol was approved by the Research Ethics Committee of the Graduate Institute of Wenzhou University with approval number WZU-2021-0328C.

**Informed Consent Statement:** Informed consent was obtained from all subjects involved in the study.

**Data Availability Statement:** Data supporting reported results in the current study can be achieved from the corrensponding author by e-mail.

**Conflicts of Interest:** The author declares no conflict of interest.

## Appendix A

### Interview question list

1.  What differences exist between this kind of learning method and your past experiences?
2.  How does virtual reality make learning different? How has it helped you?
3.  What benefits does the virtual reality learning method have as a whole?

4.     What difficulties do you have when using virtual reality technology for English reading?
5.     Please give examples of what you learned and gained the most throughout this process.
6.     How do you think the use of technology like virtual reality will help you learn new knowledge in English reading class?
7.     Would you want to learn with virtual reality again? For which particular subject(s)? Why?
8.     Would you recommend that your peers learn with the virtual reality learning method? Why do you think they need such a learning method or why would they like it?
9.     Would you recommend that your teachers teach with virtual reality? Why do you think they need such a teaching method or why would they like it?
10.    Please give suggestions for how this method can be improved.

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
