# Peer review of "Technological Solutions for Sustainable Development: Effects of a Visual Prompt Scaffolding-Based Virtual Reality Approach on EFL Learners’ Reading Comprehension, Learning Attitude, Motivation, and Anxiety"

_sustainability, doi:10.3390/su132413977_

Round 1

Reviewer 1 Report

Dear authors,

I have an excellent opportunity to review your research paper: “Technological solutions for sustainable development: Effects of 2 a visual prompt scaffolding-based virtual reality approach on 3 EFL learners’ reading comprehension, learning attitude, motivation, and anxiety”, thank you for your efforts.

I found many interesting concepts and ideas, which were put into discussion, for example: this paper presents results revealed the positive effects of the VPS-VR approach on students’ EFL reading comprehension, learning motivation, and English learning anxiety.

There are some strong advantages of this manuscript:

  • the paper is well-organized and has several sections;
  • the topic is actual and the problem is announced: it was found that experimental students’ lower-level skills of reading comprehension, such as information location and text 26 comprehension, were significantly improved, rather than the higher-level skills of reflection and 27 evaluation.
  • manuscript includes both theoretical discussion and practical case study (experiment);
  • authors present results and give some recommendations: To provide a vivid EFL learning environment, a visual prompt 18 scaffolding-based VR (VPS-VR) approach was proposed to enhance students’ reading comprehension skills. To investigate the effectiveness of the proposed approach, an experiment was conducted 20 in an English reading course at a Chinese university;
  • I agree with the main thesis and believe that the findings are relevant and interesting. Everything is logical, clear, and well-ordered.

There are some minor recommendations, which should be improved in manuscript:

  • I recommend you to include one more reference: Lee, J. Hee, Olga A. Shvetsova The impact of VR application on student's competency development: A comparative study of regular and VR engineering classes with similar competency scope/ Sustainability 2019, 11(8), 2221; https://doi.org/10.3390/su11082221 Impact Factor: 3.473 (2020); Gargrish, Shubham & Mantri, Archana & Singh, Gurjinder & Faridi, Harun. (2020). Measuring Students' Motivation towards Virtual Reality Game-Like Learning Environments. 164-169. 10.1109/Indo-TaiwanICAN48429.2020.9181362.
  • It could be interesting to mention advantages and disadvantages of chosen methods in different cases;
  • Some graphical materials (for ex., figure 1) can impress reading with improving of their quality (visualization effect) and do not forget to submit its title;
  • template of interview paper is required as Appendix;
  • Please, mention the limitations (obstacles) of your research.

Thank you one more time for this interesting contribution, I have a great pleasure to read it. Congratulations to Authors!

Author Response

Dear Reviewer,

Thank you for your comments for us to improve our reasearch quality. In order to repond in a good format, we have uploaded a file to explain our corrections. 

Reviewer 2 Report

More up-to-date articles should be added, such as Dalim, CSC, Sunar, MS, Dey, A. and Billinghurst, M. (2020). Use of augmented reality with voice input for language learning of non-native children. International Journal of Human-Computer Studies, 134, 44-64. https://www.sciencedirect.com/journal/international-journal-of-human-computer-studies/vol/134/suppl/C Herrero, JMA, Bernal, NC and Torrijos, MB (2021). An augmented reality project in the English classroom. Studies, (42). Sarmiento, BR, Prados, M. Á. H., Bernal, NC and Gómez, MCA (2021). Teacher literacy in gamification mediated by ICT. The last. Media Education, 12 (1), 53-65.

The discussion, as well as the instruments section, should be expanded, including more examples of the topics used and the reliability of all the instruments.

Also, some of the figures are not appreciated with sufficient quality.
Journal references do not include DOI. I suggest dividing the last paragraph into shorter and more direct sentences: being the conclusion, it stands as the most important and interesting part of the investigation. There are other published investigations that have worked on the use of ICT through different techniques. It is recommended that the authorship of this article contrast and / or complement its discussion with other researchers.

Author Response

Dear Reviewer, 

Thank you for you comments for us to improve our research quality. To respond in a good format, we have uploaded a file to explain our corrections

Round 2

Reviewer 1 Report

Well done, thank you! There is one recommendation left: please, structure in result section how do you answer your research questions, for example; question 1: what are the findings for this question, etc.

Author Response

Thank you for your valuable comments. Based on your comments, we have revised again our manuscript. The revised manuscript(revised text has been in red) has been uploaded.

Thank you.

We have added brief conclusions for each research questions in the Result Section to present our results clearly.

This result answered the RQ1, that is, the students learning with the VPS-VR approach improved their EFL reading comprehension more than those learning with the VR and TI approaches.

This result answered the RQ2, that is, the students learning with the VPS-VR approach improved their learning motivation in EFL reading comprehension more than those learning with the VR and TI approaches.

This result answered the RQ3, that is, the students learning with the VPS-VR approach had less learning anxiety in EFL reading comprehension learning, compared with those learning with TI approaches.

The results of the interview answered the RQ4, indicating that students learning with VPS-VR approach reported positive learning experience, such as, better learning experience and friendly learning environment, compared with those learning with the VR and TI approaches.
